

# Construction of a novel five programmed cell death-related gene signature as a promising prognostic model for triple negative breast cancer

Quanfeng Shao[1],[*], Hai-yan Gao[2],[*], Zi-ying Wang[1], Yu-ling Qian[3] and Wei-xian Chen[1],[4]

[1] Department of Breast Surgery, Changzhou No. 2 People's Hospital, The Third Affiliated Hospital of Nanjing Medical University, Changzhou, Jiangsu, China
[2] Department of Breast Surgery, Changzhou Cancer Hospital, Changzhou, Jiangsu, China
[3] Kangda Clinical College, Kangda College of Nanjing Medical University, Lianyungang, Jiangsu, China
[4] Post-doctoral Working Station, Changzhou No. 2 People's Hospital, The Third Affiliated Hospital of Nanjing Medical University, Changzhou, Jiangsu, China
[*] These authors contributed equally to this work.

Corresponding author
Wei-xian Chen,
chenweixian@njmu.edu.cn

## ABSTRACT

**Background:** Triple negative breast cancer (TNBC) is a more aggressive subtype of breast cancer that usually progresses rapidly, develops drug resistance, metastasis, and relapses, and remains a challenge for clinicians to treat. Programmed cell death (PCD), a conserved mechanism of cell suicide controlled by various pathways, contributed to carcinogenesis and cancer progression. Nevertheless, the prognostic significance of PCD-related genes in TNBC remains largely unclear, and more accurate prognostic models are urgently needed.

**Methods:** Gene expression profiles and clinical information of TNBC patients were obtained from The Cancer Genome Atlas (TCGA) and Gene Expression Omnibus (GEO) database. Least absolute shrinkage and selection operator (LASSO) and multivariate Cox regression analysis were used to establish the PCD-related gene signature. Kaplan-Meier plotter, receiver operating characteristic curves, and nomogram were applied to validate the prognostic value of the gene signature. Gene set enrichment analysis was carried out to investigate the pathways and molecular functions.

**Results:** Five PCD-related genes including SEPTIN3, SCARB1, CHML, SYNM, and COL5A3 were identified to establish the PCD-related risk score for TNBC patients. Patients stratified into high-risk or low-risk group showed significantly different survival outcome, immune infiltration, and drug susceptibility. Kaplan-Meier and receiver operating characteristic curves showed a good performance for survival prediction in different cohorts. Gene set enrichment analysis revealed that the five-gene signature was associated with tumor metabolism, cancer cell proliferation, invasion and metastasis, and tumor microenvironment. Nomogram including the five-gene signature was established.

**Conclusion:** A novel five PCD-related gene signature and nomogram could be used for prognostic prediction in TNBC. The present work might offer useful insights in

digging sensitive and effective biomarkers for TNBC prognosis prediction and establishing accurate prognostic model in clinical management.

## INTRODUCTION

Breast cancer is the most common malignancy among women worldwide and a leading cause of cancer-related death (*Giaquinto et al., 2022*). Triple negative breast cancer (TNBC) accounts for approximately 10–20% of breast cancer cases and lacks expression of the estrogen receptor, progesterone receptor, and human epidermal growth factor receptor-2. Substantial strides in regular screening, surgical resection, adjuvant chemotherapy, precise radiotherapy, and immune regulators have resulted in decreasing the 5-year mortality of TNBC over the last three decades (*Korde et al., 2021*; *Lee, 2023*). However, TNBC is a more aggressive subtype of breast cancer that usually progresses rapidly, develops drug resistance, metastasis, and relapses, and remains a challenge for clinicians to treat. These, along with the fact that TNBC represents a heterogeneous group of diverse subtypes with own biological and molecular characteristics (*Jiang et al., 2019*), highlight the need to identify more sensitive and effective prognostic biomarkers as surrogates of clinical and pathological features (*Duffy et al., 2015*).

Cell death includes accidental cell death and programmed cell death (PCD). Unlike accidental cell death caused by extreme physical, chemical, or mechanical damage, PCD is a conserved mechanism of cell suicide that is controlled by various pathways, including apoptosis, ferroptosis, cuproptosis, and so on (*Wang et al., 2024*; *Chen et al., 2024*). Apoptosis is the most extensively studied form of PCD and is morphologically characterized by cell shrinkage, organelles destruction, cytoplasmic condensation, and nuclear fragmentation (*Saraste & Pulkki, 2000*). Ferroptosis is characterized by disordered intracellular iron ion flow and significant accumulation of reactive oxygen species and lipid peroxide levels without the need for caspases (*Stockwell et al., 2017*). Cuproptosis is a new form of PCD triggered by accumulation of intracellular copper level and aggregation of mitochondrial lipoylated proteins (*Tsvetkov et al., 2022*). With the deepening of PCD research, various PCD modes have been discovered and studied, such as autophagy, necroptosis, pyroptosis, disulfidptosis, oxeiptosis, alkaliptosis, parthanatos, netotic cell death, entotic cell death, and lysosome-dependent cell death (*Chen et al., 2024*).

Mounting evidence showed that PCD contributed to carcinogenesis and cancer progression, and thus have attracted increasing attention in oncology research. Studies with regard to PCD-related gene signature as a prediction model are emerging. For example, *Chen et al. (2024)* screened 149 PCD-related differentially expressed genes, of which INHBA, LRRK2, HSP90AA1, HSPB8, and EIF2AK2 were identified as the hub genes of esophageal squamous cell carcinoma. *Cao et al. (2023)* constructed a 16 PCD-related gene model with potential in predicting prognosis and response to immune

checkpoint inhibitors in cancer. *Dong et al. (2024)* identified seven PCD-related genes to establish the PCD-related risk score for the advanced non-small cell lung cancer model, effectively stratifying overall survival in patients with advanced non-small cell lung cancer. However, the implication of PCD-related genes in TNBC is still unknown. It is therefore necessary to elucidate the molecular significance of PCD-related genes in TNBC and their correlations with survival outcomes and treatment efficacy. In this study, five PCD-related genes associated with prognosis were identified, and a risk model was constructed to assess the prognosis and accurately stratify TNBC patients to enhance survival outcomes. The findings of the present work might offer useful insights in digging sensitive and effective biomarkers for TNBC prognosis prediction and establishing accurate prognostic model in clinical management.

## MATERIALS AND METHODS

### Data collection

The transcriptomic, clinicopathological, and survival information of TNBC patients were obtained from The Cancer Genome Atlas (TCGA) database. Patients with unavailable prognostic information or incomplete gene expression data were excluded. GSE58812, GSE21653, and GSE135565 were downloaded from the Gene Expression Omnibus (GEO) database and used as validation cohorts. PCD-related genes including 580 apoptosis genes, 87 ferroptosis genes, 15 entotic cell death genes, 454 autophagy genes, nine parthanatos genes, 14 cuproptosis genes, 52 pyroptosis genes, eight netotic cell death genes, 220 lysosome-dependent cell death genes, five oxeiptosis genes, and seven alkaliptosis genes were retrieved from publications studying PCD in cancer, and a total of 1,160 PCD-related genes were finally listed after duplication elimination, according to other researchers' study protocol (*Chen et al., 2024*; *Cao et al., 2023*; *Dong et al., 2024*) (Table S1).

### Clustering and identification of PCD-related genes

In the present study, R language was used for statistical computing and graphics. Consensus clustering was performed by using the ConsensusClusterPlus package in R language. The optimal cluster "K" was obtained by using the cumulative distribution function (CDF), and number of groups was determined according to the relative CDF delta area plot stability as previously reported (*Cancer Genome Atlas Research Network, 2017*). Differentially expressed genes (DEGs) between selected PCD clusters were identified by using the limma package in R language. They were defined as follows: $|\log_2 FC| > 1.5$ and false discovery rate < 0.05.

### Cell culture and PCR

TNBC cell line MDA-MB-231 and human mammary epithelial cell line MCF-10A were purchased from Procell (Wuhan, China). MDA-MB-231 was cultured in dulbecco's modified eagle medium (DMEM; Gibco, CA, USA) supplemented with 10% fetal bovine serum (Gibco, CA, USA) and 1% penicillin-streptomycin (Gibco, CA, USA). MCF-10A was maintained in DMEM medium supplemented with 5% horse serum, 20 ng/mL epidermal growth factor, and 1% penicillin-streptomycin from Procell (Wuhan,

China). Both types of cells were cultured in a humidified incubator at 37 °C with 5% CO2. Total RNA was isolated from cultured cells by using TRIzol reagent (Invitrogen, CA, USA). Reverse transcription was performed according to the manufacturer's instructions (Vazyme, Jiangsu, China). Expressions of the selected genes were further examined by using the SYBR Green method (Vazyme, Jiangsu, China). The QuantStudioTM 5 Real-Time PCR System (Thermo Fisher, MA, USA) was employed to conduct the data analysis. Cyclic threshold (CT) ($2^{-\Delta\Delta CT}$) method was used to calculate the data. All reactions, including the negative controls, were tested in triplicate.

## Construction and validation of a prognostic model based on PCD-related genes

Differentially expressed PCD-related genes were narrowed down *via* the least absolute shrinkage and selection operator (LASSO) method by using the glmnet package in R language. Risk score for each patient was obtained according to a linear combination of expression values (weighted by the coefficient of a multivariate Cox regression analysis). *Risk Score* $= \sum_{i=1}^{n} Coefficient_i \times Expression_i$. Coefficient i stands for the coefficient of relative prognostic PCD-related genes in the multivariate Cox regression model, and expression i represents the expression level of each PCD-related genes. Based on the optimal cut-off value (−0.918997797) of the risk score, patients were divided into high-risk group and low-risk group. Likewise, patients in the GSE58812, GSE21653, and GSE135565 cohorts were also stratified into high-risk group and low-risk group for validation, followed by Kaplan-Meier analysis and receiver operating characteristic (ROC) curve analysis.

## Analysis of biological functions and pathway

Selected genes were uploaded to the GeneMANIA database to obtain protein-protein interaction information. Kyoto Encyclopedia of Genes and Genomes (KEGG) and Gene Ontology (GO) pathway enrichment analysis were carried out by using the clusterProfiler package in R language. Infiltration of immune cells in high-risk group and low-risk group was performed by using the immunocor package in R language. Seven immune infiltration algorithms including MCP-counter, xCell, CIBERSORT, CIBERSORT abs.mode, EPIC, quanTIseq, and TIMER were used to evaluate the association of five PCD-related genes and risk score with immune infiltration. Box plots were applied to show the differences in drug sensitivity between high-risk group and low-risk group by using the oncoPredict package in R language.

## Establishing and validating the predictive nomogram

Independent prognostic factors were obtained from univariate and multivariate Cox regression analysis, and a predictive nomogram was then established by using the nomoR package in R language. ROC curves were generated by using the timeroc package in R language to check the survival rates at 365 days, 1,095 days, and 1,825 days. Calibration plots of nomogram were used to describe the predicted 365 days, 1,095 days, and 1,825 days survival events and the actual observed results.

## Statistical analysis

All statistical analysis were conducted by using R software (version 4.1.0, http://www.R-project.org/) with the selected packages including limma (version 3.58.0), clusterProfiler (version 4.10.0), maxstat (version 0.7–25), pROC (version 1.18.5), ConsensusClusterPlus (version 1.68.0), glmnet (version 4.1–7), oncoPredict (version 0.2.3), nomoR (version 1.0.3), timeroc (version 0.5.1), and survival (version 3.5–7). Student's t test or Chi-square test was used to determine differences between variables. Wilcoxon test was applied to compare the proportional differences of tumor-infiltrating immune cells. Pearson correlation analysis was employed to discern relationships between distinct variables. Kaplan-Meier method was utilized for survival analysis. Univariate and multivariate Cox regression analysis were performed to obtain significant prognostic factors and their independence. ROC curve was applied to examine the accuracy of nomogram predictions. Statistical significance was assumed when $P < 0.05$.

## RESULT

### Clustering and identification of PCD-related genes

Information of 109 TNBC patients was downloaded from the TCGA database and compared with 1160 PCD-related genes from previous publications (Fig. 1). Univariate Cox regression analysis was used to screen PCD-related genes associated with survival, and a total of 29 genes (VDAC1, ATP6V0D2, DAPK2, ZNF385A, MILR1, SPTLC2, PLA2G15, LAMP3, ATP6V0D1, MT1G, LIPT1, MUL1, LGALS8, HPS6, PINK1, SVIP, CREB3L1, ACKR3, SERPINE1, LRSAM1, GPR137, BCL2A1, BCL2L10, NCK2, CTSD, CDKN1A, FZD9, RRP8, and TPCN1) were enrolled. CDF was applied to categorize the optimal number of clusters. When K was identified as 2, clustering results were relatively stable and CDF delta displayed the slowest decreasing trend, demonstrating that the differences were most significant when TNBC samples were divided into two groups (Figs. 2A–2B). As shown in the present work, samples were clustered into two groups: C1 and C2, as exhibited in the heat map (Fig. 2C). In addition, clustering consistency plots with other Ks (from 3 to 10) samples showed lower average intra-group consistency, when compared with the optimal number of clusters (Fig. 2D). Kaplan-Meier curve was used to analyze the survival difference between two groups (Fig. 2E). Subgroup C1 had remarkable better prognosis with respect to subgroup C2, indicating that survival difference could be related to the differentially expressed PCD-related genes between C1 and C2. Therefore, the volcano map and heat map of the differentially expressed PCD-related genes between C1 and C2 were drawn by using the limma package in R language (Figs. 2F–2G).

### Construction of PCD-related prognostic model for TNBC patients

According to the preliminarily obtained gene dataset, a prognostic model was constructed *via* LASSO analysis by using the glmnet package in R language. After picking the optimal penalty parameter, λ associated with the minimum five fold cross-validation, a total of five PCD-related genes, namely, SEPTIN3, SCARB1, CHML, SYNM, and COL5A3, were then chosen as candidate genes (Figs. 3A, 3B). Survival information of these genes were evaluated by using the survival package in R language. In particular, SEPTIN3, SCARB1, CHML, and

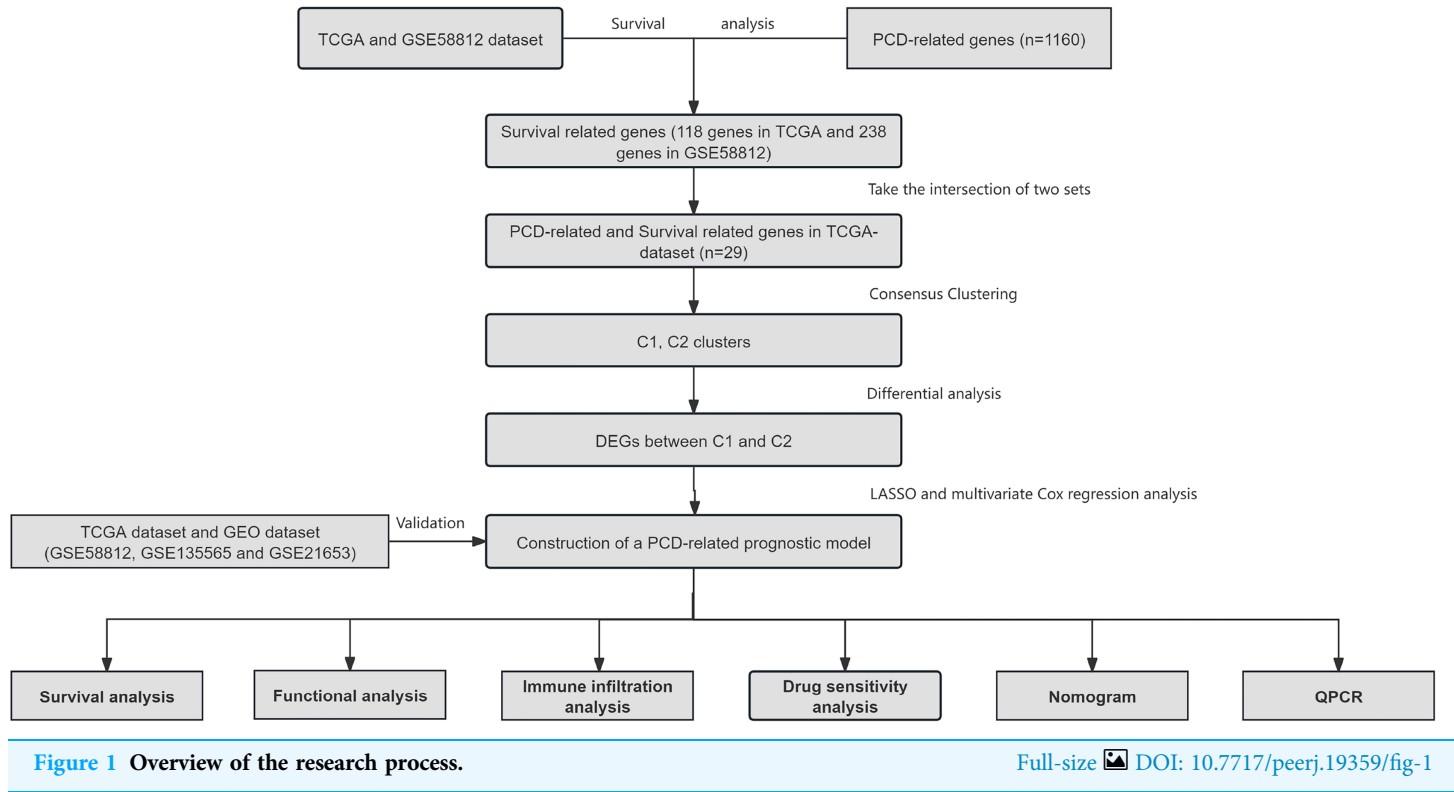

**Figure 1 Overview of the research process.**

SYNM were associated with better prognosis; whereas COL5A3 was responsible for poor prognosis (Fig. 3C). Based on the LASSO analysis and multivariate Cox regression analysis, these five PCD-related genes were finally used for prognostic model construction. Risk score for each patient was calculated as follows: risk score = −0.0799972791070521 * SCARB1 + 0.196866693004799 * COL5A3 − 0.153621818676914 * CHML − 0.141523874769071 * SEPTIN3 − 0.0623684500004447 * SYNM. Then, 109 TNBC patients in the TCGA cohort were stratified into high-risk group and low-risk group, according to the optimal cut-off value (−0.918997797) of risk score calculated by using the maxstat package in R language. When compared with the patients in the low-risk group, patients in the high-risk group showed a significantly poor survival probability (Fig. 3D). By using the pROC package in R language, time-dependent ROC analysis was carried out to further evaluate the prediction efficiency of the constructed gene signature, with the areas under curve (AUC) of 365 days, 1,095 days, and 1,825 days being 0.90, 0.89, and 0.89 (Fig. 3E), respectively. Relationship among risk score, gene expression, and survival status was also checked. Increased expressions of SCARB1, SEPTIN3, CHML, and SYNM were observed in patients from low-risk group, and elevated level of COL5A3 was found in patients from high-risk group (Figs. 3F–3G), indicating that SCARB1, SEPTIN3, CHML, and SYNM were protective factors and COL5A3 was a risk factor.

## Validation of PCD-related prognostic model for TNBC patients

To solidify the accuracy of PCD-related prognostic model, the GSE58812, GSE21653, and GSE135565 were used as validation cohorts. Patients from these datasets were therefore

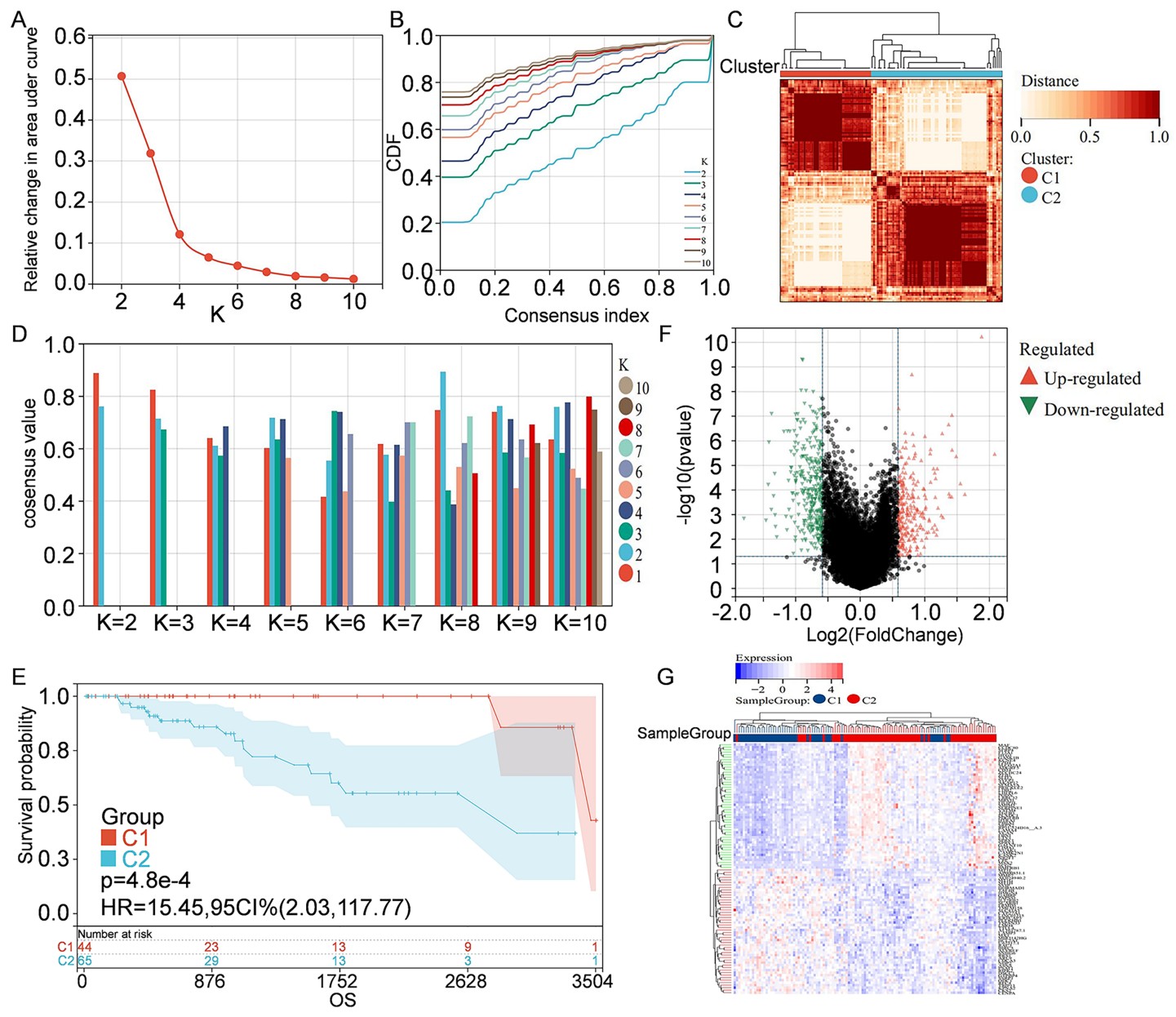

**Figure 2 Clustering and identification of PCD-related genes.** (A) Relative change in area under CDF curve. (B) CDF curves of Ks from 2 to 10. (C) Heat map showing sample clustering results, with consensus K identified as 2. (D) Clustering consistency plots for Ks from 2 to 10. (E) Kaplan—Meier curves for cluster C1 and C2. (F) Volcano map of the differentially expressed PCD-related genes between C1 and C2. Significantly up-regulated or down-regulated genes are respectively shown in red or green. (G) Heat map of the differentially expressed PCD-related genes between C1 and C2.

divided into high-risk group and low-risk group, based on the optimal cut-off value of calculated risk score as described above. Correlations among risk score, gene expression, and survival status were also evaluated. As shown in the Kaplan-Meier curves, patients in the high-risk group had a significantly worse overall survival, with respect to patients in low-risk group (Fig. 4). Moreover, patients in the low-risk group expressed increased levels of SCARB1, SEPTIN3, CHML, and SYNM; whereas patients in the high-risk group

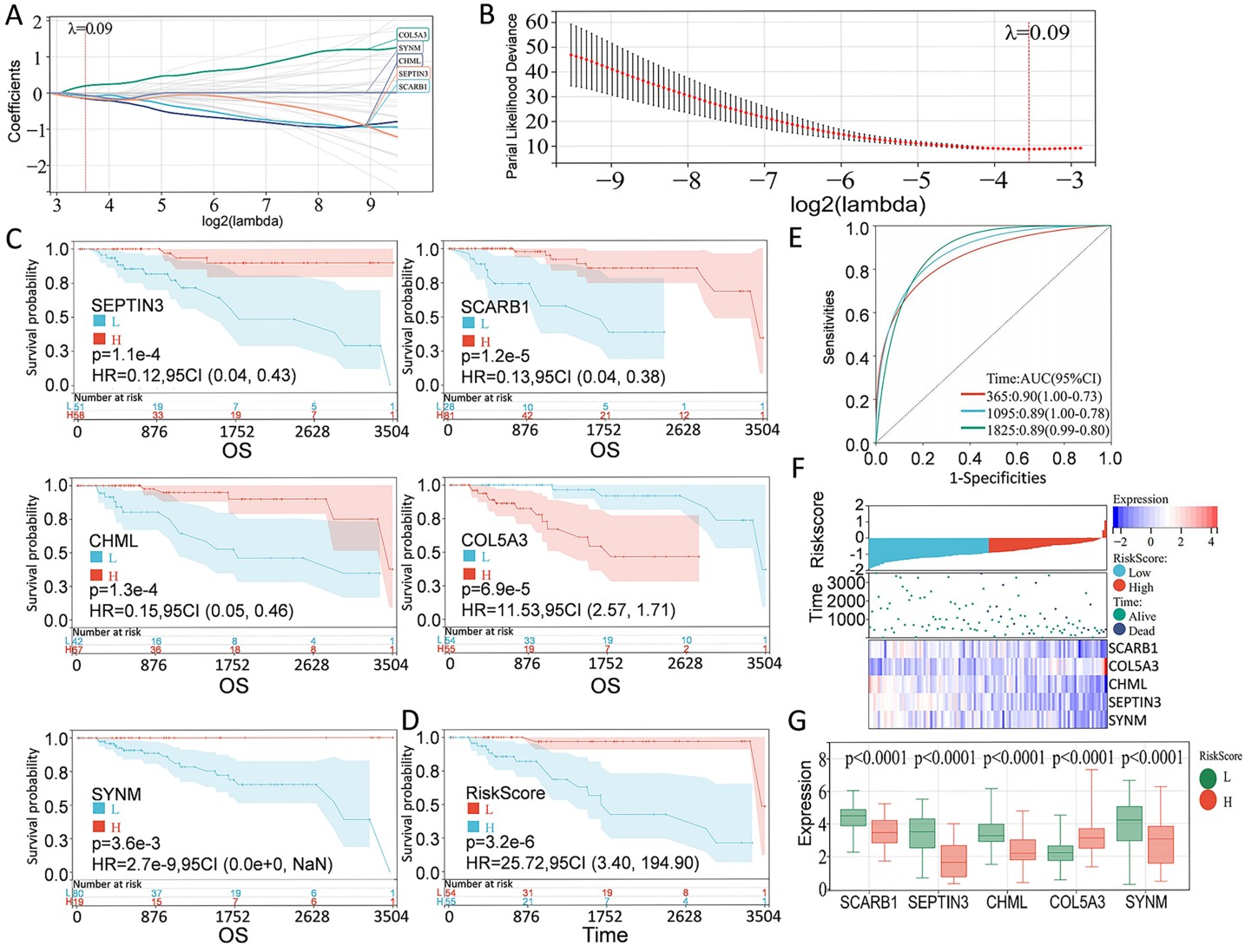

**Figure 3 Construction of PCD-related prognostic model for TNBC patients.** (A) LASSO regression analysis of five PCD-related genes associated with prognosis. (B) Optimal penalty parameter λ identified by five fold cross-validation. (C) Prognostic values of SEPTIN3, SCARB1, CHML, COL5A3, and SYNM in TNBC were analyzed. (D) Kaplan—Meier curves showing the overall survival of patients in high-risk group and low-risk group. (E) ROC curves of the five-gene signature prediction model. (F) Correlations among risk score, heat map of gene expression, and survival status of TNBC patients. (G) E xpression s of five PCD-related genes between high—risk group and low—risk group.

displayed elevated levels of COL5A3 (Fig. 4). The results collectively showed that the constructed PCD-related prognostic model was accurate for survival outcome prediction in TNBC patients.

## Elucidation of pathways and molecular functions in the TCGA cohort

To better understand the differences in signaling pathways and molecular functions between the high-risk group and the low-risk group, gene set enrichment analysis was carried out. Volcano map of the differentially expressed genes between the high-risk group and the low-risk group were drawn by using the limma package in R language (Fig. 5A).

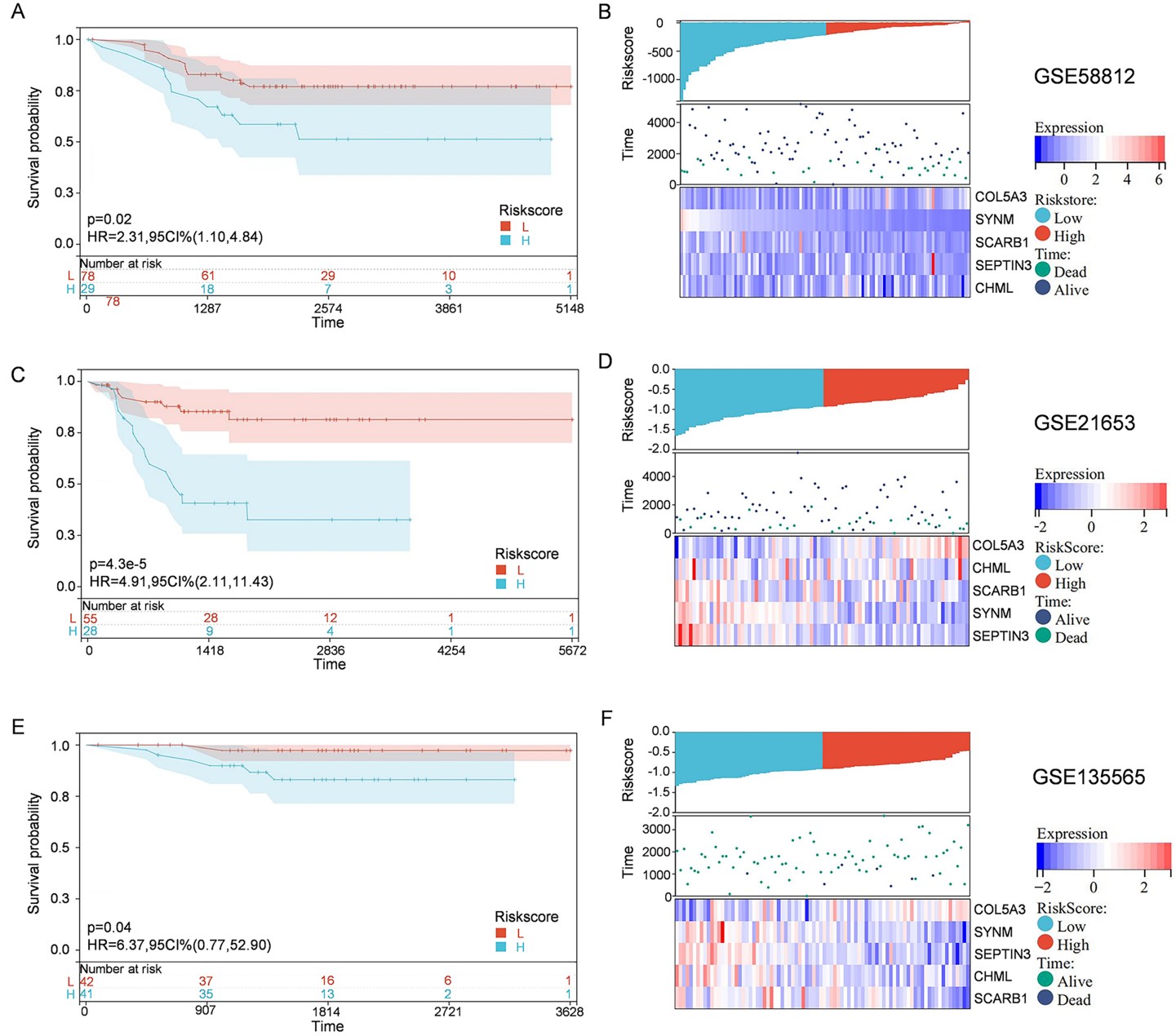

**Figure 4 External validation of the PCD-related prognostic model.** Kaplan—Meier curve s show ing the overall survival of TNBC patients in high-risk group and low-risk group in GSE58812 (A), GSE21653 (C) and GSE135565 (E). Correlations among risk score, heat map of gene expression, and survival status of TNBC patients in GSE58812 (B), GSE21653 (D) and GSE135565 (F).

KEGG analysis indicated that many metabolism-related molecular pathways were enriched, including protein and fat digestion and absorption, glycerolipid, glycerophospholipid, and arachidonic acid metabolism (Fig. 5B). Pathways related to tumor cell invasion and metastasis, such as the extracellular matrix (ECM)-receptor interaction, and relaxin signaling pathway, were also significantly enriched. GO analysis of biological processes, cell components, and molecular functions showed that the

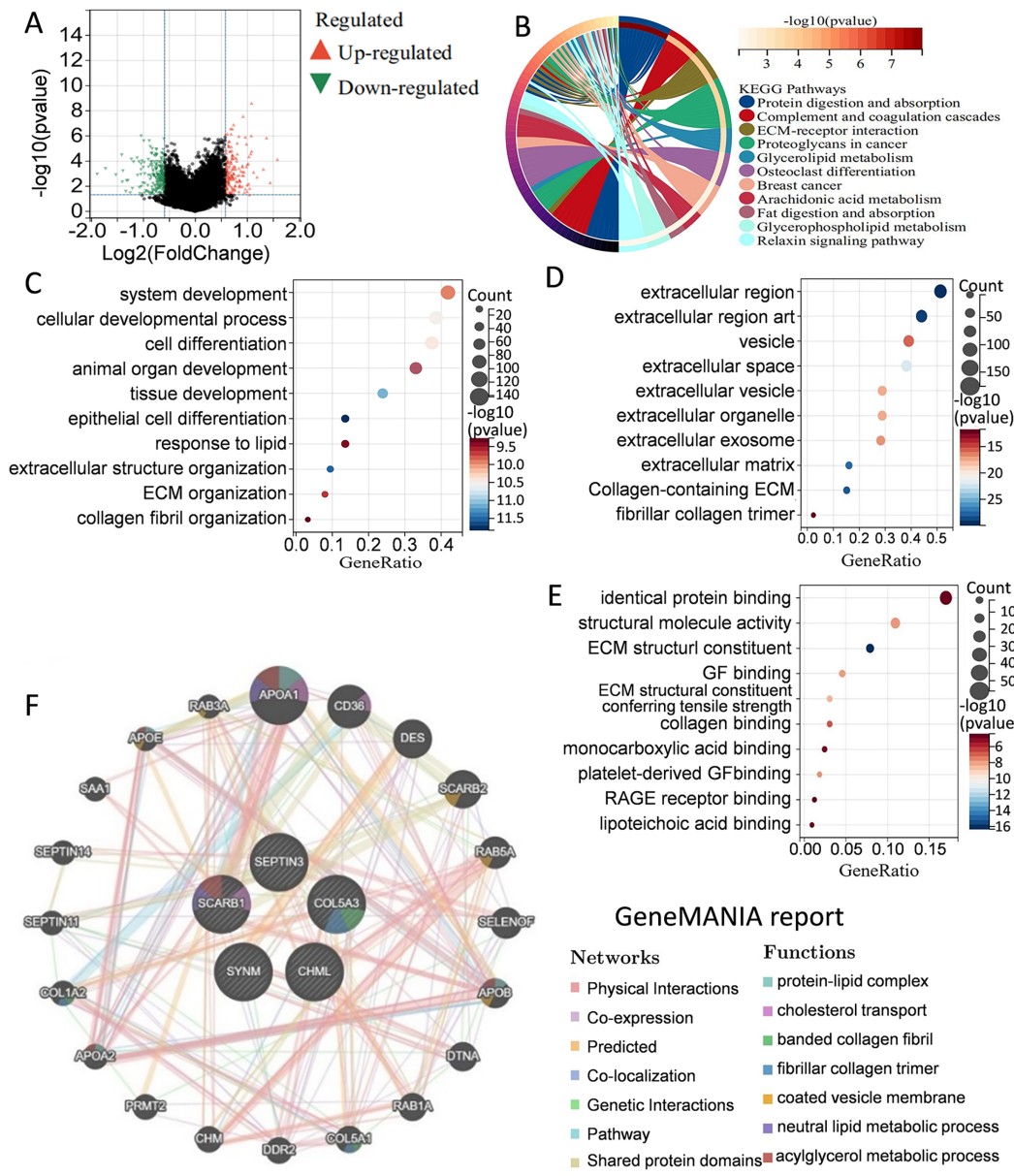

**Figure 5 Elucidation of pathways and molecular functions.** (A) Volcano map of the differentially expressed genes between high-risk group and low-risk group. (B) KEGG analysis showing many metabolism-related pathways. GO analysis showing enrichment of biological processes (C), cell components (D), and molecular functions (E). (F) Prediction and analysis of genes that were functionally similar to these five genes by using GeneMANIA database.

differentially expressed genes between the high-risk group and the low-risk group were mainly involved in cell proliferation, extracellular matrix, protein binding, and collage binding (Figs. 5C–5E). Interestingly, several cell components responsible for intercellular communication were also detected, such as extracellular region, exosomes, and extracellular vesicles. GeneMANIA database was applied to predict the genes with similar functions to these five genes. A total of 20 genes were selected, and function analysis indicated that they took part in cholesterol transport, collagen, and metabolism process

(Fig. 5F). Gene set enrichment analysis demonstrated that the high-risk group was significantly enriched in coagulation, angiogenesis, P53 pathway, apoptosis, and epithelial-mesenchymal transition (Fig. S1). Collectively, the above results showed that the differences between the high-risk group and the low-risk group classified by the PCD-related prognostic model were mainly associated with tumor metabolism, cancer cell proliferation, invasion and metastasis, and tumor microenvironment.

## Immune infiltration, and drug susceptibility analysis

Immune cells play a significant role in oncogenesis, progression, and prognosis of TNBC. CIBERSORT algorithms were further performed to explore the immune cell landscape of TNBC patients in the TCGA cohort. Analysis of immune cell infiltration patterns revealed substantial variations in the distribution of 22 distinct immune cell types in the high-risk group and the low-risk group (Fig. 6A). Such variations in the proportions of tumor-infiltrating immune cells might represent an intrinsic feature that could characterize individual differences. Moreover, the proportions of different subpopulations of tumor-infiltrating immune cells were weakly to moderately correlated (Fig. 6B). Seven immune infiltration algorithms were used to evaluate the association of five PCD-related genes and risk score with immune infiltration. The results indicated that risk score was positively correlated with immune infiltration; whereas COL5A3 and CHML were negatively correlated with immune infiltration (Figs. 6C, 6D). Box plots were applied to show the differences in drug sensitivity between the high-risk group and the low-risk group by using the oncoPredict package in R language. Low-risk group was more sensitive to most chemotherapy drugs, with respect to high-risk group, suggesting a potential factor for better prognosis in the low-risk group (Fig. 7A). Remarkably, the high-risk group was relatively sensitive to ABT737, Bl2536, and Daporinad, when compared to the low-risk group (Fig. 7B). This would provide a new way to study more effective chemotherapy regimen for high-risk TNBC patients.

## Expression levels of PCD-related genes involved in prognostic model

Expression levels of the five PCD-related genes in the TNBC cell line and the mammary epithelial cell line were analyzed by PCR. Specifically, SCARB1, SEPTIN3, CHML, and SYNM were down-regulated, while COL5A3 was significantly upregulated in MDA-MB-231 compared with that in MCF-10A (Fig. 8). These results, along with the bioinformatics analysis data, demonstrated that the prognostic model is meaningful.

## Construction of a nomogram

Univariate Cox regression analysis indicated that overall survival was significantly correlated with risk score, T, N, and stage in the TCGA cohort (Fig. 9A), and multivariate Cox regression analysis revealed that risk score and N stage were the independent prognostic factors (Fig. 9B). To build a useful predictive method, a prognostic nomogram that integrated independent prognostic factors, namely, PCD-related risk score and N stage, was generated (Fig. 9C). Based on the time-dependent ROC curves, the AUCs of nomogram achieved 0.99, 0.85, and 0.87 at 365 days, 1,095 days, and 1,825 days,

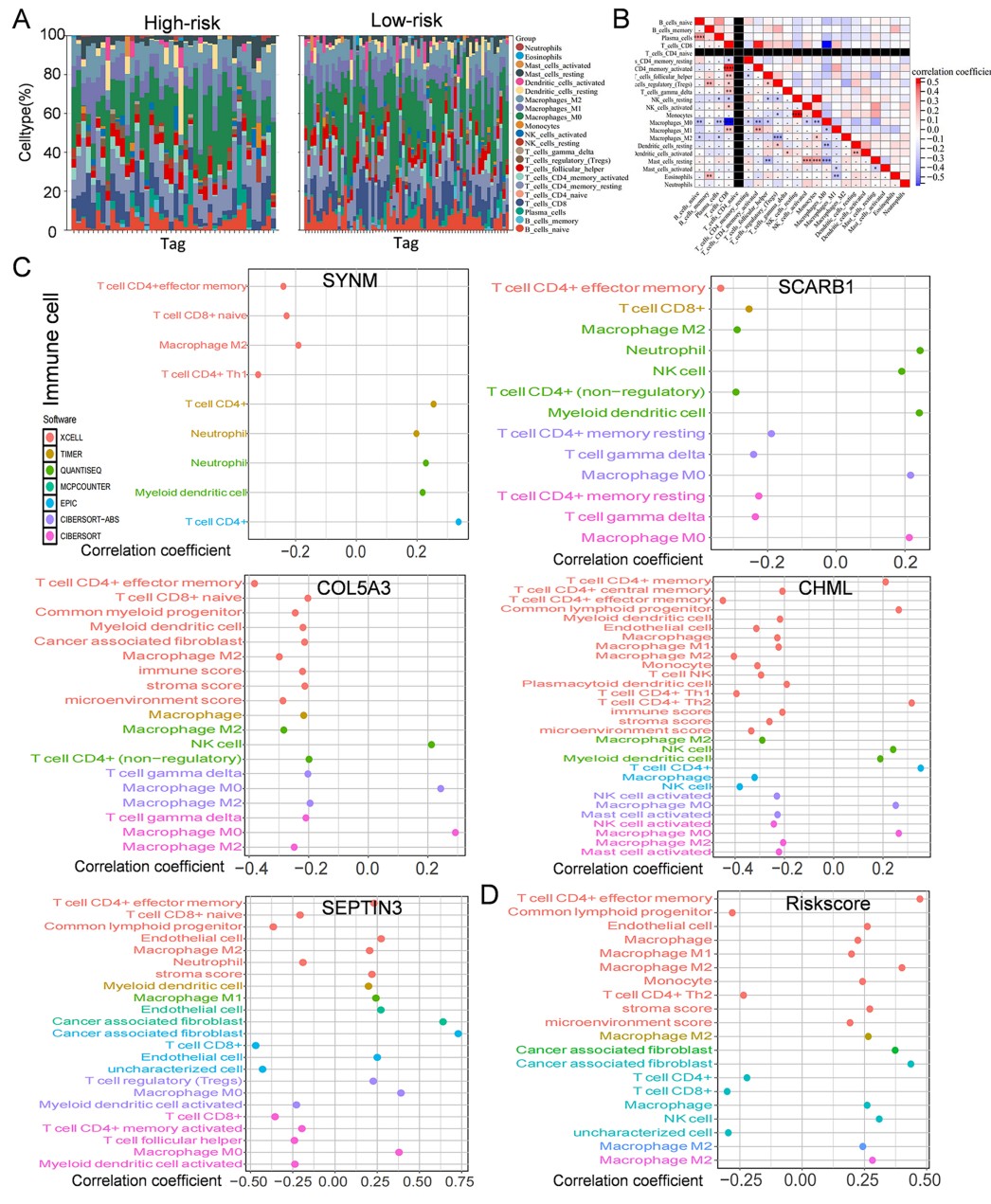

**Figure 6 Immune infiltration, and drug susceptibility analysis of the five-gene signature.** (A) Bar graph of immune cell infiltration showing distribution of 22 distinct immune cell types in high-risk group and low-risk group. (B) Immune cell correlation matrix. Correlation between SCARB1, COL5A3, CHML, SEPTIN3, SYNM (C), risk score (D) and immune infiltration in TNBC patients.

respectively (Fig. 9D). Calibration curves showed the prediction value of the nomogram and demonstrated high accuracy of the predicted survival (Fig. 9E). These results demonstrated that the nomogram could be a promising method for predicting overall survival in TNBC patients.

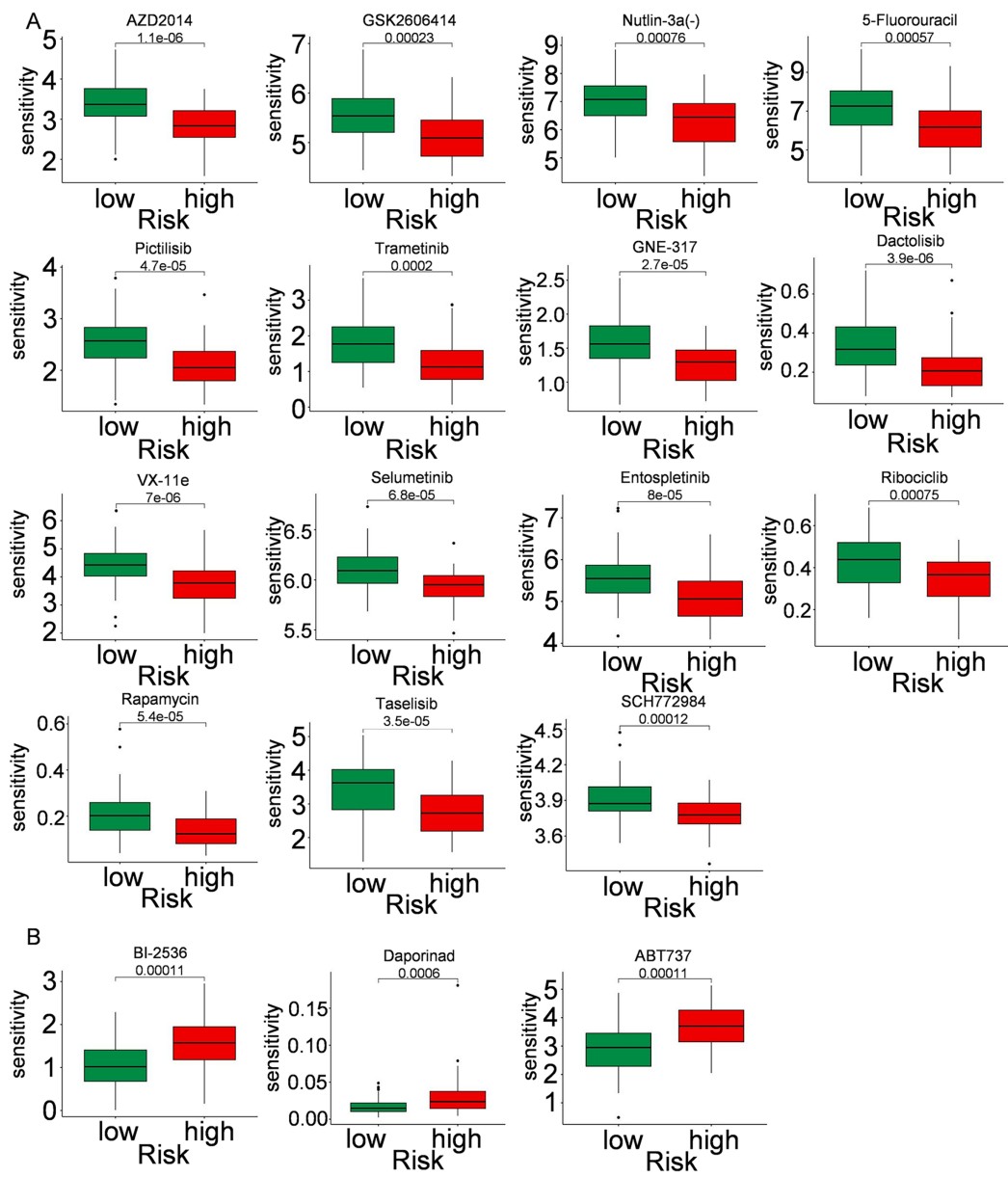

**Figure 7 Drug sensitivity in TNBC.** Box plot show ing differences in chemotherapy sensitivity between high-risk group and low-risk group.

## DISCUSSION

Breast cancer is the most common malignancy among women worldwide, and the subtype of the particular note is TNBC, which lacks expression of estrogen receptor, progesterone receptor, and human epidermal growth factor receptor-2. Both clinical and pathological features have been widely used to predict therapeutic response and long-term results. Since TNBC represents a heterogeneous group of diverse subtypes with its own biological and molecular characteristics (*Jiang et al., 2019*), identification of novel prognostic biomarkers

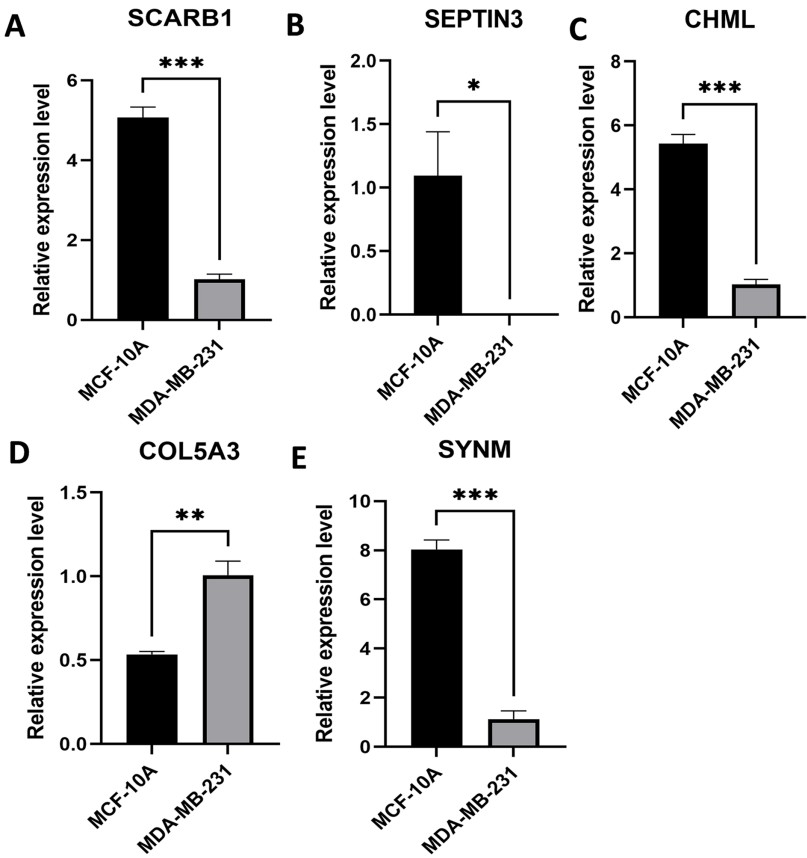

**Figure 8 Expression levels of the selected PCD-related genes.** Expression levels of SCARB1 (A), SEPTIN3 (B), CHML (C), COL5A3 (D), and SYNM (E). $^*P < 0.05$; $^{**}P < 0.01$; $^{***}P < 0.001$.

and establishment of more accurate prognostic models remain imperative and warranted in TNBC research (*Duffy et al., 2015*).

   PCD is a common gene regulated cell death mode in multicellular organisms. With the deepening of PCD research, various PCD modes are being discovered and studied (*Wang et al., 2024*; *Chen et al., 2024*). Recently, signatures based on the differentially expressed PCD-related genes have gained much attention and shown great potential in prognosis prediction of cancer (*Chen et al., 2024*; *Cao et al., 2023*; *Dong et al., 2024*; *Cancer Genome Atlas Research Network, 2017*; *Cheng et al., 2023*; *Sha et al., 2022*; *Liu et al., 2023*). *Chen et al. (2024)* screened 149 PCD-related differentially expressed genes, of which INHBA, LRRK2, HSP90AA1, HSPB8, and EIF2AK2 were identified as the hub genes of esophageal squamous cell carcinoma. *Cao et al. (2023)* constructed a 16 PCD-related gene model with potential in predicting prognosis and response to immune checkpoint inhibitors in cancer. *Dong et al. (2024)* identified seven PCD-related genes to establish the PCD-related risk score for the advanced non-small cell lung cancer model, effectively stratifying overall survival in patients with advanced non-small cell lung cancer. In the present work, a novel five-gene signature including SEPTIN3, SCARB1, CHML, SYNM, and COL5A3 was identified from a number of PCD-related genes and applied for

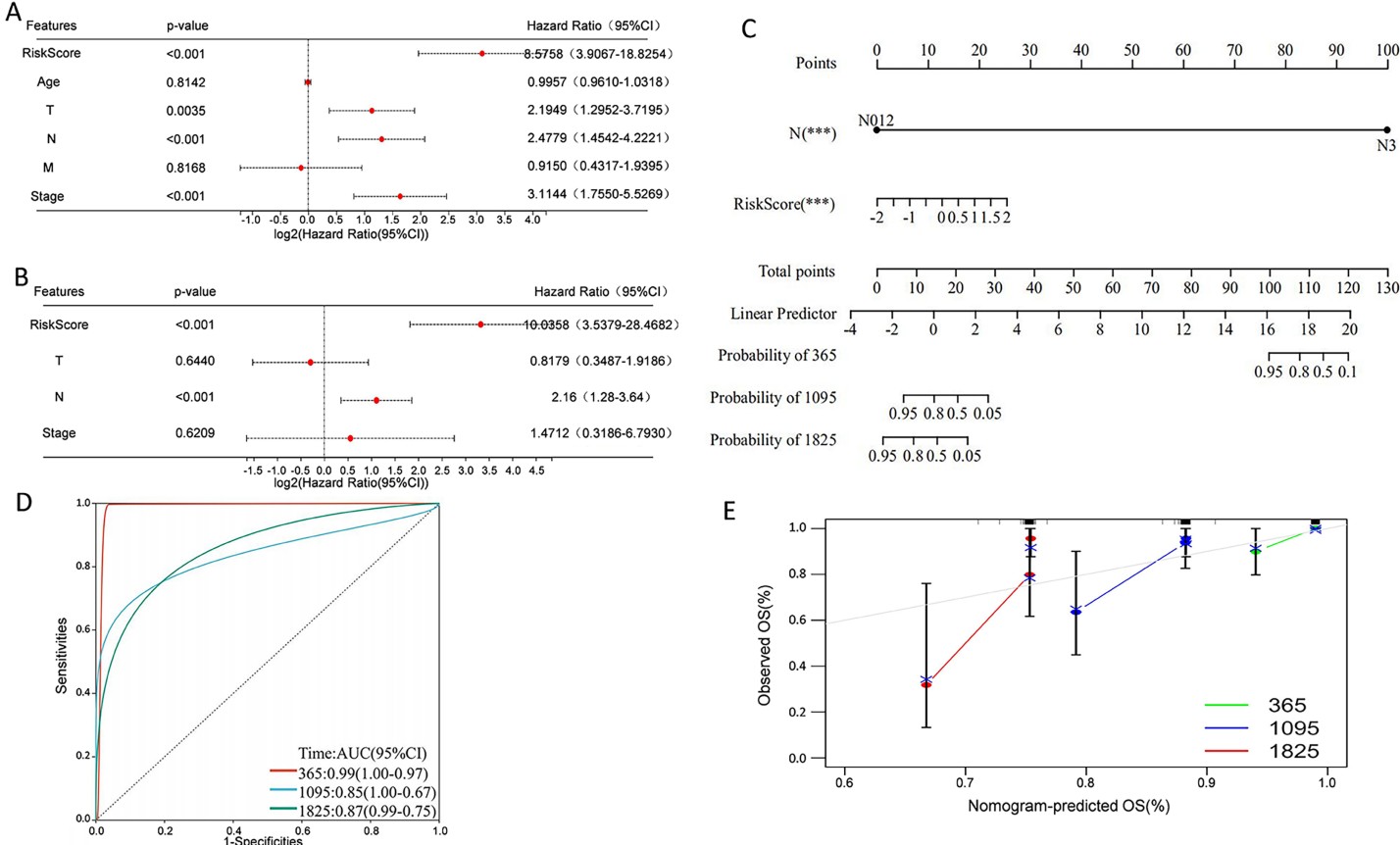

**Figure 9 Construction of a nomogram for TNBC patients based on the PCD-related risk score and clinical features.** (A) Forrest plot of the univariate Cox regression analysis. (B) Forrest plot of the multivariate Cox regression analysis. (C) Nomogram for predicting 365 days, 1,095 days, and 1,825 days overall survival of TNBC patients. (D) ROC curves for 365 days, 1,095 days, and 1,825 days overall survival of the nomogram. (E) Calibration curves for predicting 365 days, 1,095 days, and 1,825 days overall survival of TNBC patients.

prognosis prediction in TNBC patients. Specifically, SEPTIN3, SCARB1, CHML, and SYNM were associated with better prognosis; whereas COL5A3 was responsible for poor outcome. Risk score based on the five-gene signature was an independent prognostic factor of TNBC and patients in high-risk group showed a significantly worse overall survival, with respect to patients in the low-risk group. Results from survival curves and AUCs values indicated that the constructed PCD-related prognostic model was accurate for survival prediction in TNBC patients not only from the TCGA cohort but also from the GSE58812, GSE21653, and GSE135565 cohorts. A prognostic nomogram that integrated PCD-related risk score and clinical N stage was therefore generated. All these results demonstrated a good performance of PCD-related prognostic model and the prediction model could be a promising indicator for TNBC survival.

Gene *SCARB1* encodes the scavenger receptor class B type I (SR-BI) glycoprotein and regulates cholesterol exchange between cells and high-density lipoproteins (*Gutierrez-Pajares et al., 2016*). Mounting studies have reported the significant role of *SCARB1* gene and SR-BI protein in cancer proliferation and progression. In the context of breast cancer,

over-expression of SR-BI could increase high-density lipoproteins-mediated proliferation of breast cancer cells *via* PI3K/AP-1 pathway (*Cao et al., 2004*). Down-regulation of SR-BI in breast cancer cells was associated with decreased cellular cholesterol content and reduced tumor aggressiveness (*Danilo et al., 2013*). COL5A3, a member of the collagen triple helical repeat family, takes part in cell growth and migration. *Amrutkar et al. (2019)* confirmed that COL5A3 promoted chemoresistance of pancreatic cancer cells to gemcitabine treatment. COL5A3 has also been shown to participate in breast cancer brain metastasis, assess the infiltration of cancer-associated fibroblasts, and predict immune and chemotherapy responses (*Zhang et al., 2021*; *Song et al., 2024*). CHML, also known as Rab escort protein 2, is one of the key factors for Rab proteins prenylation. Researchers have found that CHML was related to the development of urothelial carcinoma, multiple myeloma, hepatocellular carcinoma, and lung cancer (*Li et al., 2008*; *Zhang et al., 2019*; *Chen et al., 2019*; *Dong et al., 2021*). CHML could promote proliferation, inhibit apoptosis, and induce metastasis of tumor cells, and high expression of CHML is associated with poor survival (*Li et al., 2008*; *Zhang et al., 2019*; *Chen et al., 2019*; *Dong et al., 2021*). SEPTIN3, a member of the septin family primarily expressed in brain and testis, is a membrane-bound presynaptic protein connected to autophagy (*Rosa et al., 2020*). *Wang, Yang & Gao (2023)* have found that SEPTIN3 was overexpressed in TNBC and was related to poor prognosis (*Yang, Wang & Gao, 2024*). Moreover, SEPTIN3 was observed to favor cell growth and oncogenesis. SEPTIN3 promoted TNBC cell aggressiveness and proliferation *via* activation of Wnt signaling pathway (*Wang, Yang & Gao, 2023*). SYNM is a type IV intermediate filament that has been reported to modulate cell adhesion and motility. Aberrant promoter methylation of gene *SYNM* was associated with lymph node metastases and advanced tumor grade (*Noetzel et al., 2010*). Besides, SYNM was positively correlated with CD8 T cells and monocytes, but was negatively correlated with γδ T cells and M1 macrophages, suggesting that SYNM could affect breast cancer cells by modulating immune-infiltrating cells (*Bao & He, 2022*). Given that only a few publications report the roles of these genes on TNBC are currently available, more thorough researches are now needed to clarify the significance and mechanism of these five genes.

Besides resisting cell death and metastasis of malignant cells, tumor metabolism and immune microenvironment are two hallmarks of cancer (*Hanahan, 2022*). In the present study, KEGG pathway and GO enrichment analysis were carried out by evaluating the differentially expressed genes between high-risk and low-risk groups classified by the five-gene signature. Several metabolism-related pathways including protein and fat digestion and absorption, glycerolipid, glycerophospholipid, and arachidonic acid metabolism were significantly enriched. Moreover, immune infiltration analysis revealed substantial variations in the distribution of 22 distinct immune cell types in the high-risk group and the low-risk group. These gene enrichment analyses collectively suggested that PCD might be closely related to tumor metabolism and immune microenvironment of TNBC.

To the best of our knowledge, the five PCD-related gene signature and nomogram have not been reported previously and could be a promising prognostic model for TNBC patients. In particular, all these five genes, namely, SEPTIN3, SCARB1, CHML, SYNM,

and COL5A3, were first applied in the prognosis prediction model for TNBC. The constructed nomogram combining the present PCD-related gene signature with patient's N stage showed improved performance in predicting overall survival, and were relatively higher than some other published nomograms for TNBC prediction (*Cheng et al., 2023*; *Sha et al., 2022*). Our study has several limitations. First, the five PCD-related gene signature and prognostic model was established by analyzing retrospective data from public databases, leading to inevitable bias. Second, few publications reporting the functions and mechanisms of these genes on TNBC are currently available. Third, TNBC usually progresses rapidly, develops metastasis and relapses, so disease-free survival, progression-free survival, and short-term survival should be included as well. Fourth, genomic and transcriptomic landscape of TNBC were quite different (*Jiang et al., 2019*). Given that TNBC subtypes were not classified in the selected TCGA and GEO database, we are unable to classify the transcriptomic, clinicopathological, and survival information of TNBC patients in the present study. Fifth, the external validity of the prognostic model and its applicability to diverse clinical settings are not evaluated, and taking the aforementioned limitations together suggest that our findings need to be validated with prospective large cohort studies, and more thorough cellular research should be performed in this field.

## CONCLUSION

In summary, this study established a new five-gene prognostic model and nomogram to predict overall survival of TNBC patients, which can not only be applied for patients management, but also provide a new direction for exploring therapeutic targets of TNBC.

### Funding

This present work was supported by grants from the Excellent Post-doctoral Program of Jiangsu Province (2022ZB820), the Changzhou Science and Technology Program (ZD202225), the Top Talent of Changzhou "The 14th Five-Year Plan" High-Level Health Talents Training Project (2022CZBJ065), the Post-doctoral Foundation of China (2022M720543 and 2019M661677), the Post-doctoral Foundation of Jiangsu Province (2019K161). The funders had no role in study design, data collection and analysis, decision to publish, or preparation of the manuscript.

### Grant Disclosures

The following grant information was disclosed by the authors:
Excellent Post-doctoral Program of Jiangsu Province: 2022ZB820.
Changzhou Science and Technology Program: ZD202225.
Top Talent of Changzhou "The 14th Five-Year Plan" High-Level Health Talents Training Project: 2022CZBJ065.
Post-doctoral Foundation of China: 2022M720543 and 2019M661677.
Post-doctoral Foundation of Jiangsu Province: 2019K161.

## Competing Interests

The authors declare that they have no competing interests.

## Author Contributions

- Quanfeng Shao conceived and designed the experiments, performed the experiments, analyzed the data, prepared figures and/or tables, authored or reviewed drafts of the article, and approved the final draft.
- Hai-yan Gao conceived and designed the experiments, analyzed the data, prepared figures and/or tables, and approved the final draft.
- Zi-ying Wang performed the experiments, prepared figures and/or tables, authored or reviewed drafts of the article, and approved the final draft.
- Yu-ling Qian performed the experiments, authored or reviewed drafts of the article, and approved the final draft.
- Wei-xian Chen conceived and designed the experiments, analyzed the data, authored or reviewed drafts of the article, and approved the final draft.

## Data Availability

Raw data is available in the Supplemental Files.

## Supplemental Information

Supplemental information for this article can be found online at http://dx.doi.org/10.7717/peerj.19359#supplemental-information.

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
