# Peer review of "Construction of a novel five programmed cell death-related gene signature as a promising prognostic model for triple negative breast cancer"

_PeerJ, doi:10.7717/peerj.19359_

## Round 0.1 · original submission · Major Revisions

Thank you for submitting your manuscript to PeerJ, which has been through the peer-review process. Reviewer comments are below. When revising your manuscript, please carefully consider all issues mentioned in the reviewers' comments: please outline every change made in response to their comments and provide suitable rebuttals for any comments not addressed. Please note that your revised submission may need to be re-reviewed.

Reviewer 1 ·

Basic reporting

1- The manuscript provides a detailed explanation of the statistical methods used for selecting the PCD-related genes, specifically the application of LASSO regression and univariate Cox regression analyses, which are well-justified. However, it would be beneficial to further clarify the initial selection criteria for the pool of 1160 PCD-related genes compiled from previous publications. Specifically, it would be helpful to understand the criteria or characteristics that made these genes relevant for inclusion in your study. Were they selected based on specific functions, pathways, known associations with programmed cell death in cancer, or other criteria?
2- The authors need to clarify the specific subtypes of TNBC included in their study, if possible. TNBC is known to be a heterogeneous group with diverse subtypes, each having its own biological and molecular characteristics.
3- Figures are presented poorly. Please upload high-quality images. It is very difficult to read and understand the legends. Please prepare high-quality images.

Experimental design

1- The introduction adequately sets up the importance of PCD in TNBC but could benefit from a more direct linkage to clinical outcomes and treatment decisions. It's recommended to briefly discuss how these PCD-related genes have been previously implicated in TNBC or similar cancers.
2- While the manuscript mentions using LASSO and Cox regression analyses, it could benefit from more detailed descriptions of the computational workflows and settings (e.g., parameters used).
3- In lines 145 and 146: “MCF-10A was maintained in special medium obtained from Procell (Wuhan, China).” What do authors mean by the special medium? Please provide medium contents.
4-In lines 162 and 163: Based on the optimal cut-off value of risk score,patients were divided into high-risk group and low-risk group.” What is the optimal cut off value?
5- Please mention the version of R packages used in M&M. For instance ConsensusClusterPlus, glmnet, and other packages in the R language
6- 2- While the manuscript mentions using LASSO and Cox regression analyses, it could benefit from more detailed descriptions of the computational workflows and settings (e.g., parameters used).

Validity of the findings

1- It is diffult tto comment as the figures have low quality.
2- 1- The discussion section reads as overly optimistic about the implications of the findings without a sufficient critique of the limitations inherent in the study. For instance, the external validity of the prognostic model and its applicability to diverse clinical settings are not adequately addressed. The manuscript would benefit from a more balanced discussion that also considers potential biases and limitations of the study.

Additional comments

1- I recommend incorporating keywords from the MeSH database to improve searchability and visibility. Such as Biomarkers and Apoptosis.
2- “Figure legends are presented twice, Please make sure that it is uploaded once
3- 2- The language of the article can be improved in some parts:
line 188 analysis -> analyses
Line 198 Information of 109 TNBC patients were downloaded ->…. Was downloaded
Line 241 To solid the accuracy -> To solidify the accuracy
Line 287 wheres -> whereas
Line 348 Researcher -> researchers
Line 361 Given that only a few publications reporting the roles -> … report the roles
Line 374 To the best of knowledge -> To the best of our knowledge

Reviewer 2 ·

Basic reporting

The manuscript would benefit from professional English editing to address minor grammatical issues and improve readability.

Provide a clearer link between PCD pathways and their potential as prognostic markers to set up the hypothesis more effectively.

Experimental design

For reproducibility, include full commands or scripts in supplementary materials or reference them explicitly.

Validity of the findings

Provide a more detailed explanation of how the findings could inform therapeutic strategies or clinical decision-making.

Clarify why the five genes (e.g., COL5A3, SYNM) are novel or significant compared to previous TNBC biomarkers.

in conclusion, briefly mention the next steps, such as prospective validation or functional studies.

---

## Round 0.2 · accepted · Accept

The authors have adequately addressed the concerns raised by the reviewers. Therefore, I recommend this manuscript for acceptance.

Reviewer 1 ·

Basic reporting

The authors have made adequate revisions, and as a result, the manuscript is now suitable for publication.

Experimental design

This content is deemed appropriate for publication.

Validity of the findings

I believe the co-authors have made sufficient revisions to make the manuscript suitable for publication, and the data is likewise suitable.